# MOES-Pred: Molecular Structural Representation Learning by Adaptive Energy-Sentinel Vibration for Generalized Property Prediction

Zhiran Hou [1]   Tinghuai Ma [1] [*]   Huan Rong [2] [*]   Li Jia [1]   Anouar Imel [3]   Heng Zhang [1]   Ming Li [1]

## Abstract

Predicting molecular properties from three-dimensional structures is fundamentally hindered by limited labeled data. While researchers have adapted self-supervised pre-training techniques from computer vision and natural language processing to address this scarcity, these approaches frequently neglect the intrinsic physical principles unique to molecular systems. From a physical perspective, denoising pre-training can be formally proven equivalent to learning molecular force fields. However, existing methods indiscriminately apply uniform noise across all molecules, thereby introducing systematic bias into the modeling of molecular distributions. To mitigate this issue, we introduce MOES-Pred, a denoising pre-training framework featuring an energy sentinel mechanism that dynamically tailors noise perturbations to individual molecules. Leveraging chemical prior knowledge, our molecule-specific noising strategies enhance conformational sampling coverage and improve distribution modeling fidelity. Extensive experiments show that MOES-Pred surpasses mainstream approaches in both force prediction and downstream quantum chemical property prediction, demonstrating substantial improvements.

## 1. Introduction

The rapid development of AI has made AI-driven Drug Discovery (AIDD) a core driver of pharmaceutical innovation(Schütt et al., 2018). Unlike traditional drug develop-ment (long cycles, high costs), AIDD accelerates target iden-tification and lead optimization via computational modeling and virtual screening. Its core lies in molecular representa-tion learning, where extracting structural and physicochemi-cal properties directly determines downstream performance. However, AIDD is constrained by data scarcity: high-quality labeled molecular data (Galson et al., 2021)(far fewer than vision/text datasets like ImageNet) is costly to obtain ex-perimentally. Thus, self-supervised pre-training(Xia et al., 2023) on unlabeled molecular data has become a key focus.

Inspired by CV/NLP pre-training, researchers migrated strategies to molecular modeling: SMILES-based meth-ods use text techniques (MLM/CLM)(Sun et al., 2020)(Hu et al., 2020), while contrastive learning constructs graph pairs. These approaches offer basic initializations but fail to fully exploit molecular properties—directly trans-planting CV/NLP methods ignores molecular-specific traits (e.g., SMILES masking misses fragment-level info) and lacks physical interpretability (only capturing statistical, not energy-based, regularities).

3D denoising pre-training addresses this gap by provid-ing a self-supervised framework with explicit physical in-terpretability. By adding noise to molecular conforma-tions and training networks to reconstruct original struc-tures, this approach is mathematically equivalent to learning molecular force fields, i.e., interatomic interaction poten-tials. Coordinate Denoising (Zaidi et al., 2023) perturbs atomic coordinates with isotropic Gaussian noise, whereas Fractional Denoising (Ni et al., 2024a) augments this ap-proach with hybrid noise applied to both dihedral angles and coordinates, thereby modeling anisotropic distributions. These physically principled formulations enable models to learn more accurate force fields and achieve superior performance in chemical property prediction. By contrast, SliDe (Ni et al., 2024b) introduces a fundamentally differ-ent noise strategy that operates directly on internal coordi-nates—specifically perturbing bond lengths, bond angles, and torsion angles—rather than Cartesian coordinates. Un-like Coord and Frad, which rely on explicit noise vector prediction to establish equivalence with force field learning, SliDe learns force fields implicitly through random slicing.

Despite its physical interpretability, 3D denoising pre-

---

[1]Department of Computer Engineering, Jiangsu Ocean Uni-versity, Lianyungang, China [2]School of Artificial Intelligence, Nanjing University of Information Science and Technology, Nan-jing, China [3]School of Information and Communication Engineer-ing,University of Electronic Science and Technology of China (UESTC), Chengdu, China. Correspondence to: Huan Rong <ronghuan@nuist.edu.cn>, Tinghuai Ma <thma@nuist.edu.cn>.

*Proceedings of the 43rd International Conference on Machine Learning*, Seoul, South Korea. PMLR 306, 2026. Copyright 2026 by the author(s).

training suffers from critical practical limitations. Existing strategies apply uniform noise distributions and magnitudes across all molecules, disregarding structural variations across molecular types and chemical environments. This introduces systematic bias: excessive noise compromises the chemical validity of rigid molecules, while insufficient noise fails to adequately explore the conformational space of flexible ones. Moreover, their energy functions are overly coarse-grained, inadequately accounting for bond/atom-type-specific interactions , which hinders the capture of complex intramolecular information. Ideally, denoising pre-training should leverage classical mechanical energy functions and adaptively adjust noise based on molecular characteristics (bond lengths, angles, dihedral distributions) to reflect real molecular physics and learn precise force fields.

To address these limitations, we propose MOES-Pred, a pre-training method customizing noise perturbation schemes per molecule. Unlike existing methods, MOES-Pred first applies diverse noise schemes within a physically reasonable range, then employs an energy sentinel to select effective schemes based on denoising outcomes. High-quality schemes are iteratively refined to replace inferior ones, enabling accurate molecular distribution modeling and improved force learning objectives. Additionally, we partition molecules into motifs via the BRICS algorithm, enhance long-range information capture by reconstructing atoms and computing intra/inter-motif interactions, and embed interatomic influence matrices into 3D conformations for denoising-based learning. Our adaptive noise strategy addresses the limitations of existing approaches and improves molecular property prediction performance.

MOES-Pred achieves superior performance on rigorous benchmarks such as MD22, ISO17, and QM9, underscoring its outstanding capability to handle large-scale complex molecular systems. This result not only validates the efficacy of our proposed approach but also highlights that accurate modeling of molecular distributions generates more optimal learning objectives—laying a solid foundation for advancing research in molecular dynamics simulations, quantum chemistry, and other related domains.

The contributions can be summarized as follows:

- We propose a pre-training framework that employs an energy sentinel mechanism to optimize molecule-specific noise perturbation schemes, enabling precise molecular distribution modeling.
- We employ the BRICS algorithm to decompose molecules into chemically meaningful motifs, compute intra-motif and inter-motif influence matrices, and embed this information into 3D conformations for denoising pre-training, effectively enhancing the model's capacity to capture long-range interactions.

- Our method achieves state-of-the-art performance on challenging benchmarks including MD22, ISO17, and QM9, demonstrating superior performance in molecular distribution modeling and downstream property prediction tasks.

## 2. Related Work

Denoising has emerged as a prominent self-supervised pre-training paradigm for molecular representation learning. Denoising-based methods have achieved SOTA performance on multiple downstream tasks, including molecular property prediction, conformational optimization, and drug screening. The core paradigm involves applying predefined noise perturbations to molecular equilibrium structures to generate noisy samples, then training neural networks to predict the added noise and reconstruct the original stable conformations. Compared to other paradigms such as masked prediction and contrastive learning, denoising-based pre-training offers a key advantage: its optimization objective is theoretically equivalent to learning molecular force fields. This equivalence endows the learned representations with both strong data fitting ability and clear physical interpretability .

Coordinate Denoising(Zaidi et al., 2023) is a seminal work that establishes the connection between denoising and force field learning. This method applies isotropic Gaussian noise to the 3D atomic coordinates of molecular equilibrium conformations and trains a graph neural network to predict the noise from perturbed inputs. Under the isotropic Gaussian perturbation assumption, the authors rigorously prove the mathematical equivalence between the denoising objective and molecular force field learning. For a sampled molecule M, the equilibrium conformation $x_0$ is perturbed according to the following distribution: $\hat{p}(x_f|x) \sim \mathcal{N}(x, \tau^2 I_{3N})$, where $x$ denotes the noisy conformation, $N$ is the number of atoms, and $I_{3N}$ is the $3N$-dimensional identity matrix. Assuming the molecular distribution follows the Boltzmann distribution constrained by $E_{\text{coord}}$, the loss function and its equivalence relation can are given by:

$$\mathbb{L}_{coord} \simeq \mathbb{E}_{p(x_f)} \big\| \text{GNN}_\theta(x_f) - \nabla_{x_f} E_{coord}(x_f) \big\|^2, \quad (1)$$

where $\text{GNN}_\theta(x_f)$ denotes a graph neural network parameterized by $\theta$, and $\simeq$ indicates equivalence of the training objectives.

To address the anisotropic distributions inherent in real molecular systems, Fractional Denoising(Ni et al., 2024a) introduces an improved noise construction strategy. Rather than using only coordinate noise, this method applies hierarchical hybrid noise to the dihedral angles of rotatable bonds and atomic coordinates, performing fractional denoising on coordinate components to balance physical interpretability with fidelity to the true molecular distribution.For a sampled

molecule M, the equilibrium conformation $x_0$ is perturbed in two steps, as shown below:

$$p(\psi_a|\psi) \sim \mathcal{N}(\psi, \sigma^2 I_m) \quad p(x_a|x) \sim \mathcal{N}(x, \tau^2 I_{3N}) \quad (2)$$

where $\psi_a$ and $\psi$ denote the dihedral angles of the perturbed and original conformations, respectively, and m is the number of rotatable bonds. Assuming the molecular distribution follows the Boltzmann distribution constrained by the energy function $E_{Frad}$, the loss function and its equivalence relation are given by:

$$\mathbb{L}_{Frad} \simeq \mathbb{E}_{p(x_f)} \left\| \text{GNN}_\theta(x_f) - \nabla_{x_f} E_{Frad}(x_f) \right\|^2. \quad (3)$$

A detailed derivation and proof are provided in Appendix A.1.

Existing denoising pre-training methods require substantial effort to carefully design specific, uniform noise distributions in order to achieve physically interpretable molecular representation learning. However, molecules also undergo small vibrations around their equilibrium positions and rotations about their center of mass. These thermal motions persist even at low temperatures, and different molecular structures exhibit varying degrees of change as temperature increases. Consequently, as illustrated in Figure 1, Coord and Frad struggle to capture the fine-grained and complex interatomic interactions within molecules. In contrast, the noise scheme proposed in this work more accurately models the physicochemical behavior of molecules, enabling finer-grained representation learning that better captures the dynamics of real physical systems.

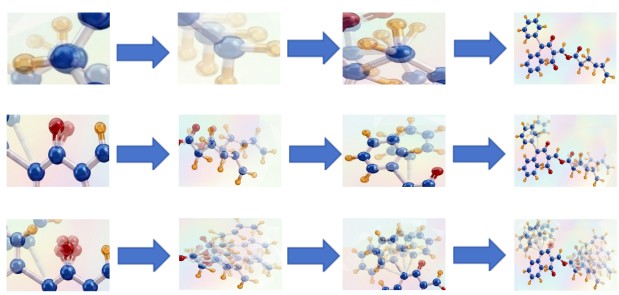

*Figure 1.* Schematic illustration of perturbations on the same molecular conformation under three different personalized noise-adding methods: (a, top) the coord method, which adds only Gaussian noise; (b, middle) the frad method, which adds personalized CAN and GCN noise; and (c, bottom) our method, with noise continuously adjusted by energy sentinels.

## 3. Our Approach

We propose a pre-training method leveraging an energy sentinel to optimize molecule-specific noise perturbation schemes, as illustrated in Figure 2.

### 3.1. Representation Structure Enhancement

We employ the BRICS (Breaking of Retrosynthetically Interesting Chemical Substructures) algorithm to systematically decompose entire molecules into distinct, chemically meaningful motifs(Inae et al., 2025). This algorithm leverages chemical domain knowledge through 16 empirically derived decomposition rules that precisely define which bonds should be cleaved within the molecular structure to generate a comprehensive set of non-overlapping substructures.The decomposition result is formally represented as:$M_G = \{V_1, V_2, \ldots, V_n\}$, a collection of $n$ distinct motifs. Each motif $M_i = (V_i, E_i)$ is characterized by its node set $V_i$ and edge set $E_i$, where $i \in \{1, 2, \ldots, n\}$.

**Definition of Influence Between Atomic Nodes** To quantify the degree of information contribution of a single atomic node to another node, we define the influence value by measuring the change in node representations: when the initial embedding of the source node $u$ is removed (set $h_u^{(0)} = \mathbf{0}$), the GNN representation of the target node $v$ will change. We use the L2 norm to measure the amplitude of this representation change, denoted as:

$$s(u, v) = \|h_v - h_{v, \neg u}\|_2 \quad (4)$$

where $h_v$ is the original GNN representation of node $v$,and $h_{v, \neg u}$ is the representation of node $v$ after removing the initial embedding of node $u$. This step directly captures whether the source node $u$ is a key information contributor to the target node $v$.

**Collective Influence of Motif on Nodes** Since the function of a molecule is usually determined by specific functional groups (i.e., Motifs), we further aggregate the atomic-level influence into the collective influence of Motifs. Let $M_j \in \mathcal{M}$ be a motif in the molecular graph $G$ with node set $V_{M_j} \subseteq V$. The collective influence of motif $M_j$ on node $v$ is defined as:

$$s_{\text{motif}}(v, M_j) = \begin{cases} \dfrac{1}{|V_{M_j}|} \displaystyle\sum_{u \in V_{M_j}} s(u, v) \\ \dfrac{1}{|V_{M_j}| - 1} \displaystyle\sum_{u \in V_{M_j} \setminus \{v\}} s(u, v) \end{cases} \quad (5)$$

When the target node $v$ belongs to the motif (i.e., $v \in V_M$), the denominator is replaced with $|V_M \setminus \{v\}|$ to exclude self-influence.

We distinguish two types of influence: (1) Intra-motif influence: It reflects the internal structural stability of a motif and measures the interactive influence among atomic nodes within the same motif; (2) Inter-motif influence: It characterizes the interactions between different motifs and quantifies

the influence of external motifs on the target atomic nodes.
**Construction of the influence matrix between atoms** Finally, we integrate intra-motif and inter-motif interactive influences to construct the inter-atomic influence matrix $\boldsymbol{S}$. Each column of the matrix corresponds to a motif, and each entry quantitatively describes the collective influence of the corresponding motif on a target atom. The formal definition is formulated as:

$$\boldsymbol{S}[v,j] = \frac{1}{|V_{M_j}|} \sum_{u \in V_{M_j}} \|h_v - h_{v,\neg u}\|_2 \qquad (6)$$

where $v$ denotes the target atomic node; $V_{M_j}$ represents the node set of the $j$-th motif; $h_v$ is the original GNN embedding of node $v$, and $h_{v,\neg u}$ denotes the optimized embedding of node $v$ after eliminating the contribution of atom $u$ within the corresponding motif. This matrix can be directly applied to downstream tasks such as molecular docking and molecular interaction prediction, and further identifies the critical functional groups that dominate molecular interaction.

## 3.2. Energy sentinel optimization of molecule-specific noise perturbation schemes

We decompose the noise into four components: stretching vibration, bending vibration, torsional vibration, and global Gaussian noise. Three binary switches $a_s, a_b, a_t \in \{0,1\}$ are introduced to control the three types of vibrational perturbations, with the constraint $a_s + a_b + a_t \geq 1$ ensuring at least one vibrational mode is activated. Following the Frad paradigm, the vibrational perturbation is formulated as:

$$x_{\text{vib}} = x_0 + a_s \cdot \Delta x_{\text{stretch}} + a_b \cdot \Delta x_{\text{bend}} + a_t \cdot \Delta x_{\text{torsion}} \quad (7)$$

where $\Delta x_{\text{stretch}}$, $\Delta x_{\text{bend}}$, and $\Delta x_{\text{torsion}}$ denote perturbations to bond lengths, bond angles, and dihedral angles, respectively, sampled within physically reasonable ranges. Isotropic Gaussian noise is then added to the vibrational perturbation, yielding the final noisy conformation: $x_{\text{noisy}} \sim \mathcal{N}(x_{\text{vib}}, \tau^2 I_{3N})$. $\Delta x_{\text{stretch}}, \Delta x_{\text{bend}}, \Delta x_{\text{torsion}}$ are the Cartesian displacements obtained by converting perturbations in internal coordinates.

During pre-training, we first generate a diverse set of noise schemes for each molecule. After the model performs denoising and reconstruction, the energy sentinel evaluates the physical plausibility of reconstructed structures and selects high-quality noise configurations. To make the energy sentinel focus not only on the overall molecular energy but also on the atoms and motifs critical to molecular physicochemical properties, we incorporate the interatomic influence matrix **s** into the energy calculation, constructing a weighted potential energy function. We define the importance weight of atom $i$ as:

$$w_i = \frac{1}{N} \sum_{j=1}^{N} \mathbf{s}[j,i] \qquad (8)$$

This weight represents the cumulative influence exerted on atom $i$ from all other atoms, reflecting its structural and functional importance within the molecule. Atoms with higher influence weights are subject to stricter supervision from the energy sentinel on their positional reconstruction errors.

Using these atomic importance weights, we construct the weighted total energy:

$$E_{\text{weighted}}(x) = \sum_{i=1}^{N} \tilde{w}_i \cdot E_i(x) \qquad (9)$$

where $E_i(x)$ is the local potential energy contribution from atom $i$. This weighting scheme can also be applied at the motif level. For motif $m$, the weight is the sum of atomic weights within the motif:

$$w_{M_j} = \sum_{i \in V_{M_j}} w_i \qquad (10)$$

The corresponding motif-level weighted energy is:

$$E_{\text{motif}}(x) = \sum_{M_j \in \mathcal{M}} \tilde{w}_{M_j} \cdot E_{M_j}(x) \qquad (11)$$

We now describe the scoring mechanism of the energy sentinel. For the $k$-th noise scheme, the model takes the noisy conformation $x_{\text{noisy}}^{(k)}$ as input and outputs the reconstructed conformation $\hat{x}^{(k)}$. We define the noise intensity of the $k$-th scheme as $\sigma_k = \|x_{\text{noisy}}^{(k)} - x_0\|_2 / \sqrt{3N}$, and the reference energy $E_{\text{ref}} = |E_{\text{weighted}}(x_0)|$ to normalize the energy term. The energy sentinel computes a score for each scheme by combining geometric reconstruction error with a physical plausibility penalty:

$$\mathcal{S}_k = \frac{1}{\sigma_k} \left( \frac{\|\hat{x}^{(k)} - x_0\|_2}{\sqrt{3N}} + \lambda \cdot \frac{E_{\text{weighted}}(\hat{x}^{(k)})}{E_{\text{ref}}} \right) \quad (12)$$

where $\lambda > 0$ is a balancing coefficient that adjusts the relative weight between the reconstruction error and the energy penalty. A lower score $\mathcal{S}_k$ indicates that the corresponding noise scheme enables accurate reconstruction while maintaining physical plausibility, representing a high-quality perturbation scheme.

**Model details** For the reconstruction loss, we employ the standard pre-training loss formulation: $\mathcal{L} = \beta \mathcal{L}_{\text{rec}} + (1 - \beta)\mathcal{L}_{\text{aux}}$, where $\beta$ is a balancing hyperparameter weighting the reconstruction loss ($\mathcal{L}_{\text{rec}}$) and auxiliary loss ($\mathcal{L}_{\text{aux}}$). We set $\beta = 0.5$ in all experiments.

To bridge atomic and motif-level representations, we introduce a motif-level feature aggregation step after the atomic embedding layer (which produces atomic embeddings $u_i, v_i$) and before the update layer (which performs

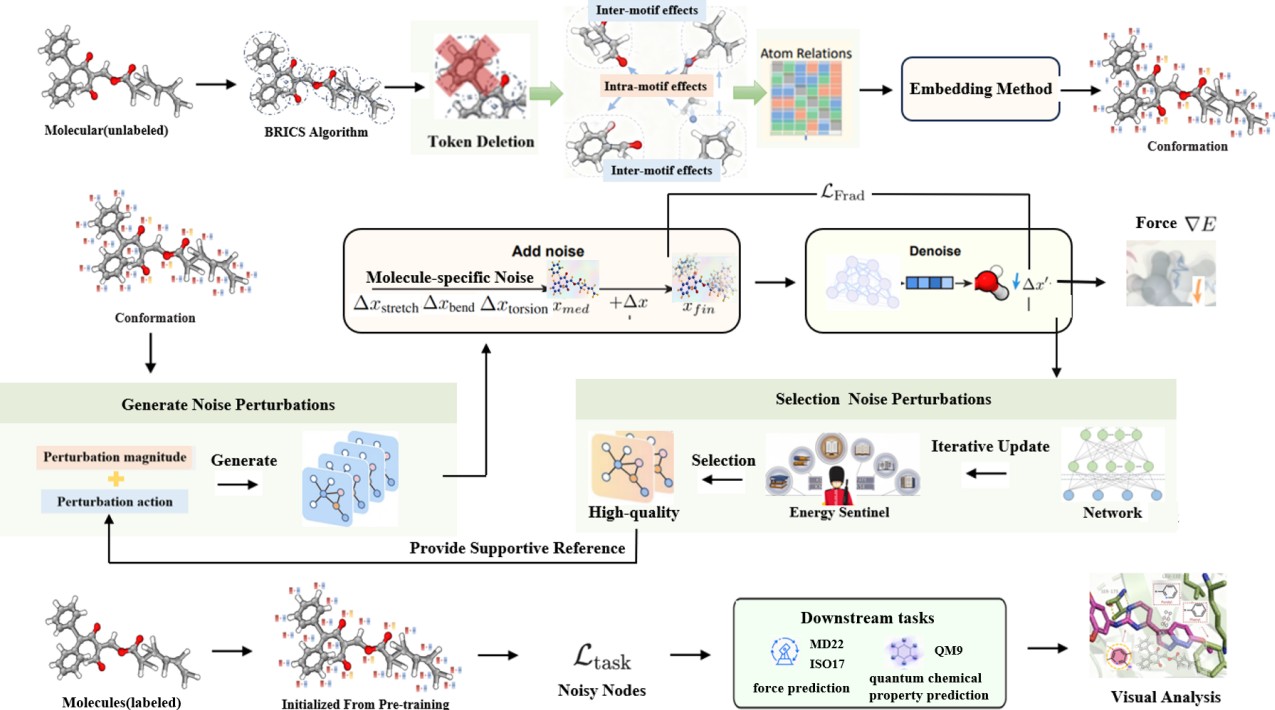

*Figure 2.* Overview of MOES-Pred. The first line illustrates the molecular representation enhancement process, the middle two lines demonstrate how the energy sentinel tailors more appropriate molecular perturbation strategies, and the last line shows the incorporation of an auxiliary loss function in labeled downstream tasks.

inter-atomic interactions). Based on the motif node attribution relationship obtained by the BRICS algorithm, we aggregate the scalar/vector features of all atoms within each motif via mean pooling to obtain motif-level representations:

$$u_m = \frac{1}{|V_m|} \sum_{i \in V_m} u_i, \quad v_m = \frac{1}{|V_m|} \sum_{i \in V_m} v_i \quad (13)$$

where $u_m$ and $v_m$ are the aggregated scalar and vector features of motif $m$, respectively, $V_m$ denotes the set of atoms in motif $m$, and $|V_m|$ is the number of atoms in this motif. This step lifts atomic-level local features to the motif level, providing a unified representation for modeling long-range interactions.

Guided by the inter-motif influence matrix $\mathbf{S}_{\text{motif}}$, we refine the aggregated motif features to capture long-range interactions, overcoming the short-range limitation of models that rely solely on local geometric features (e.g., distances, bond angles). For each motif $m$, we aggregate feature contributions from all other motifs ($m' \neq m$) via weighted summation to obtain long-range enhanced motif features:

$$u_m^{\text{long}} = \sum_{m' \neq m} \mathbf{S}_{\text{motif}}[m, m'] \cdot u_{m'} \quad (14)$$

$$v_m^{\text{long}} = \sum_{m' \neq m} \mathbf{S}_{\text{motif}}[m, m'] \cdot v_{m'} \quad (15)$$

where $\mathbf{S}_{\text{motif}}[m, m']$ serves as a long-range influence weight, ensuring that feature contributions are proportional to the actual influence intensity between motifs : the stronger the influence of motif $m'$ on motif $m$, the more significant the correction effect on the long-range features of motif $m$, achieving precise transmission of long-range information.

To incorporate motif-level long-range information into atomic representations, we propagate the refined motif features($u_m^{\text{long}}, v_m^{\text{long}}$) back to each atom contained in the motif. For atom $i$, according to its belonging motif $m(i)$ (i.e., $m(i)$ represents the motif index corresponding to atom $i$), we add the long-range motif features to the original atomic features via a residual connection:

$$u_i^{\text{new}} = u_i + \beta \cdot u_{m(i)}^{\text{long}}, \quad v_i^{\text{new}} = v_i + \beta \cdot v_{m(i)}^{\text{long}} \quad (16)$$

where $\beta \in [0, 1]$ is a learnable balancing coefficient, used to adjust the weight ratio between long-range motif features and the original local features of atoms, preventing excessive interference of long-range information with local structure modeling. The enhanced atomic features ($u_i^{\text{new}}, v_i^{\text{new}}$) after propagation are used as the input to the update layer, enabling subsequent inter-atomic attention interaction to incorporate both local geometric information and long-range cross-motif influence, significantly improving the model's ability to capture long-range information.

*Table 1.* Contrastive learning for energy (kcal/mol) and force (kcal/mol/Å) prediction tasks on MD22 & ISO17 datasets

| Datasets | Tasks | TorchMD-NET | | Coord | | F-VRN | | F-RN | | w/o B | | w/o N | | w/o E | | Our | |
|---|---|---|---|---|---|---|---|---|---|---|---|---|---|---|---|---|---|
| | | RMSE | MAE | RMSE | MAE | RMSE | MAE | RMSE | MAE | RMSE | MAE | RMSE | MAE | RMSE | MAE | RMSE | MAE |
| **MD22** | AT-AT | 0.404 | 0.294 | 0.411 | 0.295 | **0.389** | **0.276** | 0.429 | 0.352 | 0.501 | 0.302 | 0.456 | 0.368 | 0.551 | 0.365 | 0.402 | 0.285 |
| | Stack | 1.012 | 0.754 | 0.988 | 0.652 | **0.366** | **0.281** | 0.498 | 0.381 | 0.981 | 0.612 | 0.502 | 0.395 | 1.001 | 0.752 | 0.382 | 0.296 |
| | Ac-Ala3-NHMe | 0.154 | 0.119 | 0.154 | 0.116 | 0.122 | 0.095 | 0.129 | 0.106 | 0.151 | 0.132 | 0.128 | 0.101 | 0.165 | 0.142 | **0.117** | **0.089** |
| | AT-AT-CG-CG | 1.123 | 0.765 | 0.954 | 0.685 | 0.796 | 0.308 | 0.925 | 0.409 | 0.815 | 0.607 | 0.886 | 0.399 | 0.925 | 0.655 | **0.702** | **0.269** |
| | DHA | 0.202 | 0.145 | 0.182 | 0.156 | 0.136 | 0.088 | 0.136 | 0.092 | 0.124 | 0.124 | 0.139 | 0.096 | 0.155 | 0.137 | **0.128** | **0.086** |
| | Buckyball | 3.121 | 1.889 | 2.953 | 1.810 | 1.321 | 0.452 | 1.330 | 0.498 | 1.627 | 1.367 | 1.365 | 0.502 | 1.925 | 1.637 | **1.257** | **0.421** |
| | Double-walled | 4.689 | 3.024 | 3.247 | 2.261 | 1.342 | 0.786 | 1.404 | 0.872 | 2.011 | 1.458 | 1.389 | 0.862 | 2.340 | 1.658 | **1.156** | **0.705** |
| **ISO17** | Energy | 4.02 | 2.58 | 3.32 | 2.05 | 1.55 | 1.38 | 1.60 | 1.48 | 2.07 | 1.72 | 1.62 | 1.55 | 2.12 | 1.87 | **1.42** | **1.36** |
| | Force | 2.55 | 1.39 | 1.96 | 1.23 | 1.38 | 1.12 | 1.51 | 1.29 | 1.69 | 1.51 | 1.65 | 1.32 | 1.86 | 1.62 | **1.26** | **1.01** |

MOES-Pred is built upon the TorchMD-Net equivariant Transformer framework Within this backbone, we integrate an Energy Sentinel, an energy-oriented neural scoring component embedded directly in the end-to-end noise addition–denoising loop. The sentinel shares atomic and motif representations with the backbone GNN and operates as follows: after the model denoises each candidate perturbation, the sentinel monitors coordinate reconstruction errors and evaluates the physical plausibility of the reconstructed conformations through a weighted potential energy function. Rather than relying on fixed human-designed priors, the sentinel's scoring criteria, including atomic importance weights derived from the influence matrix S and the energy constraint are grounded in the GNN's learned representations. As the backbone network improves during training, the sentinel's energy constraints co-evolve accordingly: in early stages, scoring emphasizes enforcing physical rationality of molecular conformations; as training progresses, the emphasis gradually shifts toward reconstruction accuracy. This tight coupling between the sentinel's filtering strategy and the model's convergence state enables progressively more precise noise selection without manual tuning.

### 3.3. An Auxiliary Loss

Following the successful practice of Coord and Frad that employs denoising as an auxiliary task during downstream training datasets, we adopt the same strategy, which has been proven to bring significant performance improvements across various molecular benchmarks.The Noisy Nodes framework (Godwin et al., 2022) achieves these gains through two mechanisms. First, the node-level denoising loss requires the model to learn diverse node features to fit different noise targets $\varepsilon_i$, thereby alleviating over-smoothing of vertex and edge features during multi-layer message passing. Second, the denoising objective guides the network to capture physically meaningful patterns from the input distribution, facilitating effective representation learning. However, the conventional Noisy Nodes approach based on Gaussian noise prediction suffers from convergence failure in force prediction tasks.To address this, MOES-Pred applies molecule-specific noise perturbation schemes and uses decoupled inputs for denoising and downstream prediction tasks. This design enables stable convergence on the challenging MD22 and ISO17 force prediction benchmarks.

## 4. Experiments

### 4.1. Experimental Settings

This section describes the experimental setup, including benchmark datasets, baseline methods, and implementation details employed in our study. Complete details are provided in Appendices A.2 and A.3.

**Datasets** For downstream evaluation, we focus on two task categories: atomic-level force prediction and molecular-level quantum chemical property prediction. For pre-training, we use the PCQM4Mv2 dataset(Nakata & Shimazaki, 2017), which contains 3.4 million organic molecules with their equilibrium conformations.We evaluate force prediction performance on two standard benchmarks: MD22(Schütt et al., 2017) and ISO17(Chmiela et al., 2023). MD22 covers dynamic trajectories of biomacromolecules and supramolecular assemblies presenting considerable challenges in terms of model scalability, molecular flexibility, and non-local interaction modeling. ISO17 consists of 129 constitutional isomers with the molecular formula $C_7O_2H_{10}$, which share identical elemental composition but exhibit distinct spatial structures. For quantum chemical property prediction, we use the QM9(Ruddigkeit et al., 2012) benchmark, which contains approximately 134,000 stable molecules composed of C, H, O, N, and F elements, with all properties computed using density functional theory (DFT).

**Baselines** We compare our method with baselines including state-of-the-art 3D pre-training methods (**Coord**(Zaidi et al., 2023), **Frad**(Ni et al., 2024a), **Transformer-M**(Luo et al., 2023) and **3D-EMGP**(Jiao et al., 2023)) and non-pre-training methods (**TorchMD-NET**(Thölke & De Fabritiis, 2022), **SphereNet**(Liu et al., 2022), **SE(3)-DDM**(Liu et al., 2023), **PaiNN**(Schütt et al., 2021), **MABNet**(Rao et al.,

2025) and **KA-GNNs**(Li et al., 2025)). Further details on baseline methods are provided in AppendixA.3.

**Implementation details** To ensure a fair comparison with Coord and Frad, all models are pre-trained on the same dataset using an identical backbone architecture, and are implemented in PyTorch with the Adam optimizer. All noise perturbations—including stretching, bending, and torsional vibrations, as well as atomic coordinate perturbations—are computed using a unified version of RDKit(Landrum et al., 2013), a fast cheminformatics tool. Training begins with a linear learning rate warm-up, followed by decay when the validation loss plateaus. All experiments are conducted on RTX 4090 GPUs. Additional implementation details are provided in AppendixA.4

### 4.2. Results on MD22 Dataset and ISO17 Dataset

As established theoretically, the denoising objective is equivalent to learning approximate forces, which improves downstream molecular property prediction through force field learning. We therefore first evaluate force learning efficacy directly. To this end, we evaluate performance on atomic-level force prediction using two molecular dynamics benchmarks: MD22 and ISO17. As shown in Table1, our method outperforms baselines on 7 out of 9 tasks. Compared to TorchMD-NET and Coord, which share our backbone architecture, our superiority is evident, demonstrating that incorporating chemical priors into denoising enables more accurate molecular distribution modeling. Compared to the VRN and RN variants of Frad, the current state-of-the-art, our method is surpassed only on two tasks. We observe that Frad significantly outperforms Coord, and within Frad, VRN substantially exceeds RN, achieving the best results on two tasks across all baseline comparisons. Moreover, our method substantially outperforms all other baselines, underscoring the importance of modeling anisotropic molecular vibrations. This improvement stems primarily from our noise design, which tailors different noise patterns to different molecules, yielding more precise molecular distribution modeling and consequently superior force learning objectives.

### 4.3. Results on QM9 Dataset

To evaluate whether our method achieves consistently competitive results across diverse property prediction tasks, we conduct experiments on the QM9 dataset. Table5 summarizes the results for all methods. Collectively, our method outperforms all competitors and achieves the best performance across 7 out of 12 molecular property prediction benchmarks. SphereNet, which efficiently captures 3D molecular geometric features through spherical coordinate message passing, and Transformer-M, which excels at fus-

ing global information across modalities, each achieve the best results on one task. We focus our comparison on denoising pre-training methods that share our backbone architecture: TorchMD-NET and Coord. Experimental results show that TorchMD-NET only surpasses our method on a single downstream task while outperforming all other baselines, but lags behind ours on all remaining tasks. Although the Frad method demonstrates improved performance over other baselines, it still trails our method on most tasks. This highlights the effectiveness of incorporating chemical priors into the noise design for diverse property prediction tasks. These results reaffirm that improved molecular distribution modeling enhances performance across diverse property prediction tasks.

### 4.4. Ablation Study

To understand the contribution of each component, we conduct ablation studies on three key elements: representation structure enhancement, the energy sentinel, and molecule-specific perturbation schemes. Specifically, "w/o E" removes the energy sentinel, "w/o B" omits the representation enhancement module, and "w/o N" excludes tailored perturbation schemes. Tables1 and 5 present performance on force prediction and diverse downstream chemical property tasks, respectively—our complete model outperforms all ablated variants.

Furthermore, results show w/o E suffers significant performance degradation (worst on force tasks), while w/o B exhibits marked reductions (least effective on downstream chemical property tasks). Notably, removing molecule-specific perturbation schemes (w/o N) impacts force tasks more than downstream tasks, suggesting chemical prior-informed noise design is critical for atomic-level force learning, whereas representation enhancement provides limited gains here. Conversely, for downstream chemical property tasks, representation enhancement yields greater improvements than the energy sentinel. The partial leading performance of Transformer-M and SphereNet highlights the need for long-range information capture, validating the necessity of our representation enhancement module. In conclusion, all modules are essential, with relative importance varying by downstream task. More detailed decomposition on the MOES-Pred ablation study is provided in the appendix A.5

### 4.5. Visual Analysis

To further compare the noise perturbation effects across methods, we select the molecule pentyl 2-(2,2-diphenyl-4-oxoazetidin-1-yl)acetate (O=C1N(COC(=O)CCCCC)C(=O)C(N1)(c2ccccc2)c3ccccc3) for visualization. This molecule is chosen for its inclusion of typical molecular ring structures and side chains: the nitrogen atom is linked to a pentyl acetate side chain

*Table 2.* Contrastive learning on the QM9 dataset for 12 downstream tasks

| Method | $\mu$ (D) | $\alpha$ ($a_0^3$) | HOMO (meV) | LUMO (meV) | Gap (meV) | $R^2$ ($a_0^2$) | ZPVE (meV) | $U_0$ (meV) | U (meV) | H (meV) | G (meV) | $C_\nu$ (cal/(mol·K)) |
|---|---|---|---|---|---|---|---|---|---|---|---|---|
| | | | | | Molecular Property Prediction Tasks | | | | | | | |
| MABNet | 0.018 | 0.052 | 19.2 | 16.8 | 31.6 | 0.359 | 1.44 | 6.81 | 6.34 | 6.81 | 7.46 | 0.026 |
| PaiNN | 0.012 | 0.045 | 27.6 | 20.4 | 45.7 | 0.070 | 1.28 | 5.85 | 5.83 | 5.98 | 7.35 | 0.024 |
| KA-GNNs | 0.028 | 0.046 | 24.6 | 18.6 | 33.5 | 0.356 | 1.39 | 6.21 | 6.35 | 6.71 | 7.56 | 0.027 |
| SphereNet | 0.030 | 0.051 | 23.4 | 18.2 | 33.6 | 0.299 | **1.11** | 6.38 | 7.51 | 6.52 | 7.94 | 0.022 |
| TorchMD-NET | 0.012 | 0.063 | 20.2 | 18.9 | 35.6 | **0.035** | 1.82 | 6.31 | 6.39 | 6.12 | 7.89 | 0.028 |
| Transformer-M | 0.037 | 0.041 | 17.5 | 16.2 | **27.4** | 0.075 | 1.18 | 9.37 | 9.41 | 9.39 | 9.63 | 0.022 |
| SE(3)-DDM | 0.015 | 0.046 | 23.5 | 19.5 | 40.2 | 0.122 | 1.31 | 6.92 | 6.99 | 7.09 | 7.65 | 0.024 |
| Coord | 0.013 | 0.052 | 17.1 | 15.2 | 32.6 | 0.462 | 1.73 | 6.62 | 6.03 | 6.89 | 6.72 | **0.021** |
| 3D-EMGP | 0.020 | 0.059 | 22.0 | 18.9 | 37.6 | 0.099 | 1.41 | 8.34 | 8.60 | 8.92 | 9.01 | 0.027 |
| F-RN | **0.011** | 0.042 | 15.4 | **14.0** | 27.9 | 0.352 | 1.42 | 5.46 | 5.68 | **5.58** | 6.33 | 0.022 |
| F-VRN | 0.012 | 0.045 | 17.3 | 14.2 | 28.6 | 0.357 | 1.42 | 5.39 | 6.48 | 6.21 | 6.45 | 0.022 |
| Ours | **0.011** | **0.040** | 15.2 | 14.2 | 27.7 | 0.344 | 1.43 | **5.38** | **5.31** | 5.69 | **6.01** | **0.021** |
| w/o N | 0.018 | 0.048 | 18.2 | 15.5 | 35.8 | 0.498 | 1.51 | 6.20 | 5.91 | 6.98 | 7.33 | 0.029 |
| w/o B | 0.026 | 0.062 | 21.2 | 14.9 | 39.8 | 0.831 | 1.68 | 6.88 | 8.08 | 8.62 | 9.51 | 0.032 |
| w/o E | 0.019 | 0.051 | 17.1 | 16.5 | 32.8 | 0.502 | 1.55 | 6.92 | 7.19 | 7.11 | 8.32 | 0.029 |

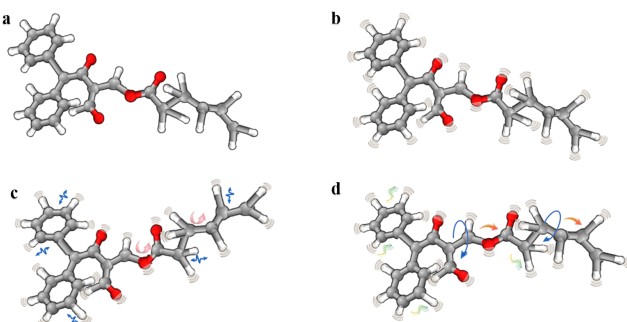

*Figure 3.* Conformation visualization. a: original noise-free conformation; b: conformation with Coord noise; c: conformation with Frad noise; d: conformation with MOES-Pred noise.

($CH_2OC(=O)CCCCC$), while the quaternary carbon on the ring is substituted with two phenyl groups (c2ccccc2, c3ccccc3). These phenyl groups are aromatic (rigid) rings, which are typically excluded during noise addition to avoid disrupting conjugation or inducing ring strain. Consequently, noise can only be applied to the pentyl acetate side chain. To more clearly highlight the differences between the three noise addition methods, we proportionally scaled up the final noise amplitude. Figure 3 illustrates the amplified molecular perturbation effects. As shown in Figure 3, Coord's noise addition results in small, disordered perturbation amplitudes due to its independent atomic perturbation strategy. To preserve chemically valid structures, the noise amplitude must be drastically reduced to prevent substructure distortion—and experiments confirm that

increasing Coord's noise amplitude significantly degrades prediction performance. In contrast, both Frad and our method achieve substantially larger amplitudes. However, Frad employs fixed-magnitude random perturbations, which inevitably generate unreasonable perturbations (see the figure). Thus, its amplitude must also be moderately reduced to balance overall noise effects. Our method, by contrast, addresses this limitation through molecule-specific amplitude design.

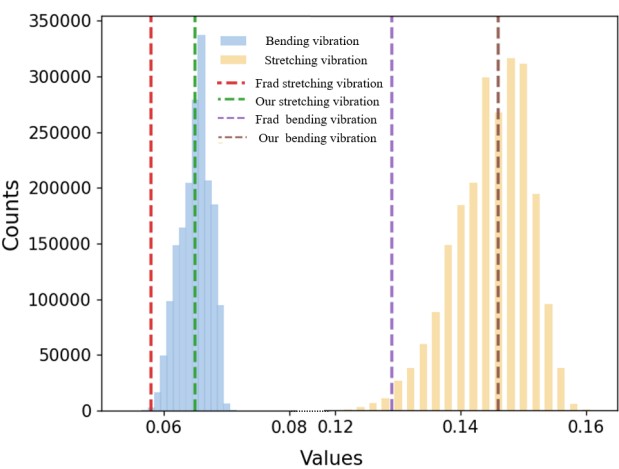

*Figure 4.* Perturbation Amplitude Statistics for Molecules.

To verify this advantage, we conducted a statistical comparison of perturbation amplitudes with Frad (using data from its original appendix). As shown in Figure Figure 4, our

method achieves superior average amplitudes for stretching, bending, and torsional vibrations compared to Frad, with particularly pronounced improvements in stretching and bending (12.6% and 13.4%, respectively). This trend likely arises because adjusting stretching/bending vibrations has a relatively weak impact on molecular energy, whereas torsional adjustments exert a more significant effect. This indicates that the Frad method reduces perturbation amplitudes to reconcile the effects of perturbations at the atomic level, within individual molecules, and across the entire molecular dataset. In contrast, our molecule-aware perturbation design effectively increases the overall perturbation magnitude. A larger noise amplitude expands the coverage of major low-energy molecular conformations, which further improves the learning efficiency of molecular force fields.

## 5. Conclusion

In this work, we presented MOES-Pred, an adaptive denoising pre-training framework that leverages an energy sentinel for molecule-specific noise optimization. This framework automatically generates and selects molecule-specific noise perturbation schemes, enabling precise modeling of molecular distributions. We further introduced motif decomposition based on the BRICS algorithm and integrate intra-motif and inter-motif influence matrices to enhance the capture of long-range interactions. Extensive experiments on MD22, ISO17, and QM9 demonstrate that MOES-Pred achieves state-of-the-art results. These results validate the effectiveness of both adaptive noise optimization and motif-aware representation learning. Overall, this work highlights that physically grounded, adaptive denoising can substantially elevate the quality of molecular pre-training representations.

MOES-Pred primarily focuses on molecular property prediction tasks utilizing 3D structural data as input and cannot accommodate other data modalities such as SMILES strings or molecular graphs. Future work may explore integration with complementary pre-training approaches to develop models capable of handling multi-modal molecular data.

## Impact Statement

Our work aims to combine chemical prior knowledge with existing data to facilitate the development of more affordable, safer, and more effective drugs. Through continuous optimization of quantum chemical property prediction methods, we expect to accelerate the discovery process of novel drugs, thereby making a substantive contribution to societal well-being.

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

# A. Appendix

## A.1. Theoretical proofs

### Proof of Equivalence Between Denoising and Force Field Learning(Ni et al., 2024a)

To simplify notations, we define: - $x_{\text{eq}} \in \mathbb{R}^{3N}$: Equilibrium conformation of the molecule, where $N$ is the number of atoms; - $x_a \in \mathbb{R}^{3N}$: Intermediate noisy conformation (obtained by adding noise to dihedral angles); - $x \in \mathbb{R}^{3N}$: Final noisy conformation (obtained by adding noise to coordinates); - $\psi_{\text{eq}}, \psi_a$: Dihedral angles of rotatable bonds in the equilibrium and intermediate conformations, respectively, with $m$ being the number of rotatable bonds; - $\simeq$: Denotes equivalent optimization objectives for the parameters $\theta$ of the Graph Neural Network (GNN) (differing only by constants or coefficients independent of $\theta$); - Boltzmann distribution: The molecular conformation distribution is related to energy by $p(x) \propto \exp(-E_{\text{Frad}}(x))$, where $E_{\text{Frad}}(x)$ is the molecular energy function corresponding to fractional denoising.

**Preliminary Assumptions** 1. The conformation distribution is a mixture distribution centered at the equilibrium state:

$$p(x) = \int p(x|x_a)p(x_a|x_{\text{eq}})p(x_{\text{eq}})dx_{\text{eq}},$$

where: - Dihedral angle noise: $p(\psi_a|\psi_{\text{eq}}) \sim \mathcal{N}(\psi_{\text{eq}}, \sigma_f^2 I_m)$ ($\sigma_f^2$ is the variance of dihedral angle noise); - Coordinate noise: $p(x|x_a) \sim \mathcal{N}(x_a, \tau_f^2 I_{3N})$ ($\tau_f^2$ is the variance of coordinate noise, and $I_k$ is the $k \times k$ identity matrix).

2. GNN model: $\text{GNN}_\theta(x)$ takes the final noisy conformation $x$ as input and outputs node-level noise predictions.

**Core Theorem** Fractional denoising loss is an equivalent optimization objective to atomic force field regression, i.e.:

$$\mathcal{L}_{\text{Frad}} = \mathbb{E}_{p(x|x_a)p(x_a|x_{\text{eq}})p(x_{\text{eq}})}\|\text{GNN}_\theta(x) - (x - x_a)\|^2 \simeq \mathbb{E}_{p(x)}\|\text{GNN}_\theta(x) - (-\nabla_x E_{\text{Frad}}(x))\|^2.$$

**Step 1: Objective Transformation (Relating Energy and Score via Boltzmann Distribution)** From the Boltzmann distribution $p(x) \propto \exp(-E_{\text{Frad}}(x))$, take the logarithm of both sides and differentiate with respect to $x$:

$$\nabla_x \log p(x) = -\frac{1}{kT}\nabla_x E_{\text{Frad}}(x),$$

where $kT$ is the product of the Boltzmann constant and temperature (a constant). Rearranging gives:

$$-\nabla_x E_{\text{Frad}}(x) = kT \cdot \nabla_x \log p(x).$$

Since $kT$ is a constant, it can be absorbed by scaling the GNN output (optimization objectives remain equivalent). Thus, we only need to prove:

$$\mathcal{L}_{\text{Frad}} \simeq \mathbb{E}_{p(x)}\|\text{GNN}_\theta(x) - \nabla_x \log p(x)\|^2. \tag{1}$$

**Step 2: Equivalence of Conditional Score Matching (Using Lemma from Vincent (2011))** According to the lemma on the equivalence of conditional score matching and marginal score matching (Vincent, 2011): for any conditional conformation distribution $p(x|x_a)$, we have:

$$\mathbb{E}_{p(x)}\|\text{GNN}_\theta(x) - \nabla_x \log p(x)\|^2 = \mathbb{E}_{p(x|x_a)p(x_a)}\|\text{GNN}_\theta(x) - \nabla_x \log p(x|x_a)\|^2 + T,$$

where $T$ is a constant independent of the GNN parameter $\theta$ (does not affect optimization equivalence).

Since $p(x_a) = \int p(x_a|x_{\text{eq}})p(x_{\text{eq}})dx_{\text{eq}}$, using the property of integral interchange, the expectation can be extended to:

$$\mathbb{E}_{p(x|x_a)p(x_a|x_{\text{eq}})p(x_{\text{eq}})}\|\text{GNN}_\theta(x) - \nabla_x \log p(x|x_a)\|^2 + T. \tag{2}$$

Combining equations (1) and (2), we only need to further prove:

$$\mathcal{L}_{\text{Frad}} \simeq \mathbb{E}_{p(x|x_a)p(x_a|x_{\text{eq}})p(x_{\text{eq}})}\|\text{GNN}_\theta(x) - \nabla_x \log p(x|x_a)\|^2. \tag{3}$$

**Step 3: Score Calculation for Gaussian Distribution** From the preliminary assumptions, coordinate noise satisfies $p(x|x_a) \sim \mathcal{N}(x_a, \tau_f^2 I_{3N})$, and its probability density function is:

$$p(x|x_a) = \frac{1}{(2\pi\tau_f^2)^{3N/2}} \exp\left(-\frac{\|x - x_a\|^2}{2\tau_f^2}\right).$$

Differentiate $\log p(x|x_a)$ with respect to $x$ (score function):

$$\nabla_x \log p(x|x_a) = -\frac{x - x_a}{\tau_f^2}. \tag{4}$$

**Step 4: Verification of Loss Function Equivalence** Substitute equation (4) into the right-hand side of equation (3):

$$\mathbb{E}_{p(x|x_a)p(x_a|x_{\text{eq}})p(x_{\text{eq}})} \left\| \text{GNN}_\theta(x) - \left(-\frac{x - x_a}{\tau_f^2}\right) \right\|^2 = \mathbb{E} \left\| \text{GNN}_\theta(x) + \frac{x - x_a}{\tau_f^2} \right\|^2.$$

Define a new GNN model $\text{GNN}_\theta'(x) = \tau_f^2 \cdot \text{GNN}_\theta(x)$ (only scaling the output, equivalent to the original model in optimization), then the above equation can be rewritten as:

$$\frac{1}{\tau_f^4} \mathbb{E} \left\| \text{GNN}_\theta'(x) + (x - x_a) \right\|^2.$$

Since $\frac{1}{\tau_f^4}$ is a positive constant factor that does not affect the position of the minimum value of the optimization objective, we have:

$$\mathbb{E} \left\| \text{GNN}_\theta(x) - \nabla_x \log p(x|x_a) \right\|^2 \simeq \mathbb{E} \left\| \text{GNN}_\theta(x) - (x - x_a) \right\|^2 = \mathcal{L}_{\text{Frad}}.$$

**Step 5: Integrating Conclusions** Combining the derivations from Steps 1 to 4, through the energy-score association of the Boltzmann distribution, equivalence of conditional score matching, score calculation for Gaussian distribution, and absorption of constant factors, we finally obtain:

$$\mathcal{L}_{\text{Frad}} \simeq \mathbb{E}_{p(x)} \| \text{GNN}_\theta(x) - (-\nabla_x E_{\text{Frad}}(x)) \|^2.$$

That is, fractional denoising loss is an equivalent optimization objective to atomic force field regression, and the proof is completed.

### A.2. Datasets

Detailed information on the datasets used in this study is presented in Table 3, and the data are available via the following links:

PCQM4Mv2: https://ogb.stanford.edu/docs/lsc/pcqm4mv2;

QM9 : https://ffgshare.com/collections/Quantum chemistry structures and properties of 134 kilo molecules/978904;

MD22 : http://www.sgdml.org/#datasets

ISO17 : http://quantum-machine.org/datasets;

### A.2.1. PCQM4Mv2

PCQM4Mv2(Nakata & Shimazaki, 2017) is a large-scale quantum chemistry dataset affiliated with the Open Graph Benchmark (OGB) , derived from the PubChemQC project, serving as a pivotal benchmark dataset in the interdisciplinary field of graph machine learning and quantum chemistry. The core task is predicting the HOMO-LUMO gap calculated via the Density Functional Theory (DFT, B3LYP/6-31G*) on the basis of the 2D graph structures of molecules. This molecular property plays a crucial role in such practical scenarios as the research and development of organic photovoltaic materials and virtual drug screening. After the SMILES-based update, the dataset contains 3,746,619 valid molecules, split into training, validation, test-dev, and test-challenge sets at a 90:2:4:4 ratio. The splitting strategy based on PubChem compound IDs

effectively prevents label leakage in model training. As the core dataset for the competition track, PCQM4Mv2 is frequently adopted to evaluate the performance of various models including graph neural networks and Transformer architectures. Notably, its distinctive feature of being independent of 3D molecular structures makes it an ideal benchmark for testing the generalization ability of graph models, and it has been widely applied to the pre-training and fine-tuning tasks of molecular property prediction models.

### A.2.2. MD22

MD22(Schütt et al., 2017) is a molecular dynamics benchmark dataset , specifically designed to evaluate the equivalent force prediction capability of machine learning force field models on complex macromolecular systems. The dataset covers four categories of biomolecules and supramolecules, ranging from a 42-atom short peptide to a 370-atom double-walled carbon nanotube. The system scale far exceeds traditional MD17, presenting new challenges in molecular size, flexibility, and non-local interactions. All trajectories are sampled at high temperatures of 400–500 K with 1 femtosecond resolution, and both potential energy and atomic forces are calculated using the high-precision PBE+MBD electronic structure method to ensure data reliability. In equivalent force validation scenarios, MD22 is used to test whether pretrained models can accurately predict atomic forces in large-scale molecules, verifying their ability to capture the physical consistency of potential energy surfaces. The dataset is commonly partitioned into training, validation and test sets via random split or the original sGDML split. Molecules in the test set are distributed differently from those in the training set in chemical space, which demands strong generalization performance from the model. MD22 has become the gold standard for measuring the performance of new-generation equivariant graph neural networks in handling realistic biomacromolecular systems and is of great significance for developing robust force field models applicable to drug design and materials science.

### A.2.3. ISO17

ISO17(Chmiela et al., 2023) is a specially designed molecular dynamics dataset focusing on equivalent force field validation tasks for isomer systems. The dataset contains 129 isomers of $C_7O_2H_{10}$, with the unique feature that all molecules have identical atomic compositions but fundamentally different chemical structures, providing an ideal platform for rigorously testing model sensitivity to structural diversity. Each isomer is equipped with complete molecular dynamics trajectories, enabling researchers to evaluate model capabilities in extrapolating across chemical space. The dataset employs a challenging splitting strategy: conformations of 80% of the molecules are used for training and validation, while all conformations of the remaining 20% unseen molecules serve as the test set, ensuring the test set covers a different chemical space from the training and validation sets, which aligns well with real-world drug screening scenarios. In equivalent force validation applications, ISO17 is used to test whether pre-trained models can accurately predict atomic forces for various isomers, assessing whether the learned chemical prior knowledge can generalize to entirely new topological structures. This dataset effectively measures a model's ability to distinguish subtle structural differences and plays an irreplaceable role in verifying chemical awareness and robustness. It has become one of the standard benchmarks for evaluating structural generalization performance in molecular force field research.

### A.2.4. QM9

QM9(Ruddigkeit et al., 2012)is the most influential benchmark dataset in the field of quantum chemistry, widely used for evaluating downstream molecular property prediction tasks. The dataset contains 134,000 stable small organic molecules composed of carbon, hydrogen, oxygen, nitrogen, and fluorine. All molecules possess DFT-optimized equilibrium geometries and are annotated with 12 key quantum chemical properties, including dipole moment, polarizability, HOMO/LUMO energy levels, energy gap, zero-point vibrational energy, internal energy, enthalpy, free energy, and heat capacity. These properties are calculated at the high-precision B3LYP/6-31G(2df,p) theoretical level, providing reliable training targets for machine learning models. The standard dataset split consists of 110,000 training samples, 10,000 validation samples, and 10,831 test samples. In downstream task applications, QM9 is used to evaluate a model's ability to predict electronic structure and thermodynamic properties from three-dimensional molecular structures. The 12 regression tasks cover different physicochemical dimensions, requiring models to simultaneously capture both geometric information and electronic structure features of molecules. QM9 has not only advanced the application of graph neural networks in chemistry but also provides an ideal fine-tuning and testing platform for pre-training methods, serving as the gold standard for measuring model generalization capability and physical consistency.

*Table 3.* Datasets statistics description.

| Dataset | Purpose | Molecule types | # Data Scale | # conformations | Calculation Methods | # atoms |
|---------|---------|----------------|--------------|-----------------|--------------------|---------|
| PCQM4Mv2 | Pre-train | Medium-sized organic molecules | 3378606 | 1 per molecule | DFT(B3LYP/6-31G*) | $\approx 30$ |
| QM9 | chemical property | Up to 9 heavy atoms, containing only C, H, O, N, F elements | 130,831 | 1 per molecule | DFT(B3LYP/6-31G*) | 3-29 |
| ISO17 | Force | Isomers of $C_7O_2H_{10}$ | 129 | 5000 per molecule | DFT(PBE0/def2-TZVP) | 19 |
| MD22 | Force | Large molecular systems (e.g., tetrasaccharide stachyose) | 22 | $\approx 4000$ per molecule | DFT(PBE+MBD) | 42-370 |

## A.3. Baselines

### A.3.1. PRE-TRAINING METHODS

**Coord**(Zaidi et al., 2023): A classic molecular denoising pre-training method that uses coordinate Gaussian noise as the core perturbation strategy, only capturing small-scale vibrations of molecular equilibrium conformations. Frequently paired with TorchMD-NET, it enables basic force learning interpretation but suffers from a single noise distribution, failing to cover the diversity of actual low-energy molecular conformations and thus having limited generalization ability.

**Frad**(Ni et al., 2024a): A molecular pre-training framework based on fractional denoising, integrating chemically aware noise (CAN) and coordinate Gaussian noise (CGN) to decouple noise design from force learning constraints. It demonstrates equivalence to atomic force learning, captures a broader range of molecular conformation distributions, and outperforms Coord comprehensively in force prediction, quantum chemical property prediction, and binding affinity tasks, with outstanding generalization capability.

**Transformer-M**(Luo et al., 2023): A cross-modal molecular representation model based on the Transformer architecture, featuring separate 2D and 3D encoding channels to adaptively process molecular 2D graph or 3D spatial structure data. Corresponding channels are activated when specific input formats are provided, fusing atomic features with structural information from different dimensions. It maintains excellent performance in both 2D and 3D molecular property prediction tasks.

**3D-EMGP**(Jiao et al., 2023): A pre-training model for 3D molecules, often used as a baseline for molecular denoising pre-training and sharing the same type of backbone as Coord and Frad. Focusing on feature extraction and denoising recovery of 3D molecular conformations, it provides basic representations for downstream molecular property prediction. However, it lacks integration of chemical priors, resulting in lower prediction accuracy than Frad for complex molecular systems.

### A.3.2. NON-PRE-TRAINING METHODS

**TorchMD-NET**(Thölke & De Fabritiis, 2022): A lightweight and efficient neural network potential model that can be implemented as a PyTorch module for GPU acceleration and integrated into mainstream molecular dynamics simulation codes. It extracts 3D molecular equivariant features and is often used as the backbone for molecular pre-training frameworks. Compatible with atomic force and energy prediction tasks on datasets such as MD17 and MD22, it exhibits excellent training and inference efficiency.

**SphereNet**(Liu et al., 2022): A 3D molecular learning model proposing a spherical message-passing mechanism based on spherical coordinates. It extracts geometric features such as interatomic distances, polar angles, and azimuthal angles to achieve translationally and rotationally invariant predictions. Significantly reducing computational complexity, it efficiently handles large-scale molecules and delivers performance comparable to full message-passing schemes on datasets like QM9 and MD17, while reducing runtime by 4x.

**SE(3)-DDM**(Liu et al., 2023): A diffusion model based on the SE(3)-equivariant state space model, adopting deterministic denoising steps instead of stochastic reverse updates to significantly improve inference efficiency. It unifies deterministic and stochastic diffusion through a reverse transfer kernel framework, effectively capturing long-range dependencies in molecular graphs. With strong generalization to large molecules, it is suitable for high-precision 3D molecular generation tasks.

**PaiNN**(Schütt et al., 2021): A polarizable atomic interaction neural network and the first molecular modeling method where equivariant models outperform invariant models in parameter tuning. Fusing scalar and vector feature embeddings, it efficiently captures interatomic distances, directions, and angle information, reducing the computational complexity of angle calculations from $O(nk^2)$ to $O(nk)$. It can distinguish similar molecular conformations and accurately describe molecular interactions such as charge transfer and polarization.

**KA-GNNs**(Li et al., 2025): A molecular feature learning model based on graph neural networks, integrating graph attention and kernel attention mechanisms to focus on local and global feature extraction of molecular structures. It efficiently captures correlative features between atoms and chemical bonds, adapting to property prediction and representation learning of small molecules. As a classic method in molecular graph modeling, it is often used as a baseline for various pre-training models.

**MABNet**(Rao et al., 2025) A molecular prediction model that first realizes direct modeling of four-body interactions, designing a four-body multi-head attention mechanism to capture high-order quantum effects. Adopting an E(3)-equivariant message-passing architecture to ensure molecular spatial symmetry, it reduces computational complexity from $O(N^4)$ to $O(N^2 \cdot E)$ through spatial sparsification and quantum-inspired pruning. It achieves significantly reduced force prediction errors on datasets like MD22 and is suitable for large molecular systems.

### A.4. Experimental settings

We present a novel pre-training framework that incorporates an improved fractional denoising (Frad) approach with an energy-sentinel update mechanism. During early training stages, the sentinel prioritizes enforcing the physical plausibility of molecular conformations; as optimization progresses, the emphasis gradually shifts toward reconstruction accuracy. This tight coupling between the sentinel's filtering strategy and the model's convergence state enables progressively more precise noise selection without manual tuning. The implementation logic of MOES-Pred pre-training is detailed in the pseudocode of Algorithms 1 and 2.

---

**Algorithm 1** MOES-Pred denoising pre-training

---

**Require:**
    $\Delta x_{\text{stretch}}, \Delta x_{\text{bend}}, \Delta x_{\text{torsion}}$ : Scale of noise
    $\Delta x \sim \mathcal{N}\left(0, \tau^2 I\right)$
    $a_s, a_b, a_t$ : Noise switch
    $\Delta x_{\text{Coord}}$: Scale of coordinate Gaussian noise
    X: Pre-training dataset of molecular conformations
    T: Training steps
**while** T $\neq$ 0 **do**
  $x_e$ = dataloader(X)                        *// random sample a molecular conformation*
  **if** $a_s = 1$ **then**
    Perturb the stretching coordinates with Gaussian noise.           *// Add Gaussian noise*
  **end if**
  **if** $a_b = 1$ **then**
    Perturb the bending coordinates with Gaussian noise.
  **end if**
  **if** $a_t = 1$ **then**
    Perturb the torsional coordinates with Gaussian noise.
  **end if**
  $x_f = x_m ed + \Delta x_{\text{Coord}}$
  $\mathcal{L} = \|\Delta x^{\text{pred}} - \Delta x\|_2^2$
  Optimise(L)                            *// Update model parameters*
  $T \leftarrow T - 1$
  Optimizing the noise scheme via Energy-Sentinel.
**end while**

---

The data preprocessing follows a funnel-shaped cascading pipeline that progressively filters samples across four dimensions: chemistry, geometry, numerics, and distribution. First, molecular topologies are parsed with RDKit to exclude chemically

---

**Algorithm 2** Energy-Sentinel Iterative Update Strategy in MOES-Pred

---

**for** iteration **do**

    **Evaluate:** Calculate energy sentinel scores $S_K$ for all schemes

    **Rank:** Sort schemes in ascending order by $S_K$ (the lower the better)

    **Truncation selection:**

        Retain: $P_{\text{elite}} = \{\, p \mid \text{rank}(p) \leq K/2 \,\}$                      *// top 50%,lowest scores as best*

        Discard: $P_{\text{discard}} = \{\, p \mid \text{rank}(p) > K/2 \,\}$            *// bottom 50%, highest scores as worst*

    **Local sampling** (40%):

    **for** each $p \in P_{\text{elite}}$ **do**

        Sample new scheme $p'$ from the parameter neighborhood $N(p)$ Collect $p'$ into $P_{\text{new-local}}$

    **end for**

    **Global random** (10%):

    Randomly sample $P_{\text{new-global}}$ from the global space

    **Recombination:** $P = P_{\text{elite}} \cup P_{\text{new-local}} \cup P_{\text{new-global}}$

    **Output:** Final high-score scheme pool

**end for**

---

*Table 4.* Hyperparameters for pre-training.

| Parameter | Value or description |
|---|---|
| Train Dataset | PCQM4MV2 |
| Data Splitting | sets at a 90:2:4:4 ratio |
| Batch size | 70 |
| Optimizer | SiLU |
| activation | AdamW |
| Warm up steps | 10000 |
| Learning rate decay policy | Cosine |
| Network structure | 256-8-32 |
| std. of bond lengths exploration step size | 0.001 |
| std. of bond angles exploration step size | 0.002 |
| std. of torsion angles exploration step size | 0.004 |
| std. of Gaussian noise exploration step size | 0.001 |

invalid instances, such as those that fail parsing or exhibit valence and aromaticity inconsistencies. Next, geometrically malformed conformations are screened out by detecting atomic clashes, extreme deviations in bond lengths and angles, or missing 3D coordinates. Concurrently, numerical outliers are removed, including entries with NaN DFT energies, anomalous HOMO–LUMO gaps, or energy values that statistically deviate from their population distribution. During the denoising pre-training stage, noise-perturbed conformations undergo further physicochemical validation; variants that suffer topological dissociation, bond-length collapse, or energy explosion upon perturbation are discarded. Finally, deduplication via canonicalized InChI retains the highest-quality representative. This pipeline ultimately yields high-quality data that is chemically valid, geometrically sound, numerically reliable, and distributionally balanced.

We present the details about the hyper-parameters of our experiments in Table 4.We adopt a unified pre-training and task-specific fine-tuning paradigm: during pre-training, the model is configured with $hidden_channels = 256 and num_layers = 8$; for downstream fine-tuning on MD22 and ISO17, we adjust the architecture from the pre-trained configuration ($hidden_channels = 256$, $num_layers = 8$) to $hidden_channels = 128$ and $num_layers = 6$, initializing the model weights from the pre-trained checkpoint with dimension truncation, without introducing additional linear projection layers to maintain lightweight efficiency.

## A.5. More detailed decomposition on Ablation Study of MOES-Pred

We appreciate this suggestion. Note that the isotropic Gaussian noise component cannot be removed, as it is required for the theoretical equivalence between the denoising objective and force field learning. We use: w/o S (removes stretching noise), w/o Bend (removes bending noise), w/o T (removes torsional noise); w/o W (removes the importance-weighting scheme in the energy sentinel).

*Table 5.* Ablation Study of MOES-Pred on the QM9 dataset for 12 downstream tasks

| Method | \multicolumn{12}{c}{Molecular Property Prediction Tasks} | | | | | | | | | | | |
|---|---|---|---|---|---|---|---|---|---|---|---|---|
| | $\mu$ (D) | $\alpha$ ($a_0^3$) | HOMO (meV) | LUMO (meV) | Gap (meV) | $R^2$ ($a_0^2$) | ZPVE (meV) | $U_0$ (meV) | U (meV) | H (meV) | G (meV) | $C_\nu$ (cal/(mol·K)) |
| w/o S | 0.013 | 0.045 | 15.8 | 15.1 | 28.5 | 0.392 | 1.48 | 5.77 | 5.63 | 6.08 | 6.41 | 0.022 |
| w/o Bend | 0.012 | 0.045 | 16.1 | 15.0 | 28.6 | 0.386 | 1.46 | 5.83 | 5.50 | 5.93 | 6.36 | 0.022 |
| w/o T | 0.013 | 0.047 | 16.4 | 14.8 | 29.3 | 0.399 | 1.45 | 5.89 | 5.53 | 6.12 | 6.55 | 0.023 |
| w/o S | 0.018 | 0.055 | 16.3 | 16.1 | 31.6 | 0.488 | 1.50 | 6.02 | 6.82 | 6.76 | 7.89 | 0.026 |
| MOES-Pred | **0.011** | **0.040** | **15.2** | **14.2** | **27.7** | **0.344** | **1.43** | **5.38** | **5.31** | **5.69** | **6.01** | **0.021** |

The results reveal several informative patterns. First, removing the importance-weighting scheme from the energy sentinel (w/o W) leads to substantial performance degradation, as the model can no longer enforce stricter supervision on positions that are structurally and functionally critical. Second, among the three vibrational noise types, removing torsional noise (w/o T) yields a slightly larger performance drop than removing bending (w/o Bend) or stretching noise (w/o S), although the differences among the three are relatively modest. This aligns with the theoretical understanding that torsional rotations constitute the primary driver of conformational change in flexible molecules; consequently, their removal constrains the model's capacity to explore conformational space. Finally, removing stretching (w/o S) or bending noise (w/o Bend) primarily impairs local structural fidelity, with comparatively limited impact on global conformational coverage.

