# OpenReview forum: "MOES-Pred: Molecular Structural Representation Learning by  Adaptive Energy-Sentinel Vibration for Generalized Property Prediction"
_ICML.cc/2026/Conference — ICML 2026 regular_

### Official Review · Reviewer_LJSQ · 2026-03-07

**Soundness:** 2
**Presentation:** 2
**Significance:** 3
**Originality:** 3
**Overall Recommendation:** 4
**Confidence:** 4

**Summary:**

In this work, the authors propose MOES-Pred, a pre-training framework designed to improve molecular property prediction through structural representation learning. The key idea is to apply diverse noise perturbation schemes within a physically reasonable range and employ an energy sentinel mechanism to adaptively select effective schemes based on the denoising outcomes.

**Compliance With Llm Reviewing Policy:**

Affirmed.

**Final Justification:**

The authors provided a thorough and well-structured rebuttal that addresses key concerns, including clarifying the backbone architecture and hyperparameters, improving reproducibility, and supporting results with additional statistical and computational analyses. While the performance gains are moderate in some cases, they are consistent across benchmarks and supported by further evidence, strengthening confidence in the approach. Overall, the framework is technically sound, and is likely to be of interest to the community, so I maintain my weak accept recommendation.

**Key Questions For Authors:**

* The paper mentions using a Graph Neural Network (GNN) as the backbone for the proposed pre-training framework, but the specific 3D GNN architecture used during pre-training is not clearly specified. Could the authors clarify which backbone model is used (e.g., PaiNN, SE(3)-Transformer, TorchMD-Net, etc.), along with the architectural configuration and hyperparameters? Providing these details would improve reproducibility and enable fair comparison with prior work.
* In Table 2, the performance differences between MOES-Pred and competing methods for some properties (e.g., HOMO, $\mu$, and $C_{v}$) appear to be relatively small. Could the authors provide statistical significance analysis, such as results averaged over multiple runs with standard deviation or confidence intervals, to verify that the improvements are not due to randomness? Additionally, it would improve readability if the second-best results were underlined in Tables 1 and 2.
* Since MOES-Pred relies on large-scale pre-training on the PCQM4Mv2 dataset, could the authors provide more details regarding the computational cost of the pre-training stage, such as training time, number of GPUs used, and total computational budget (e.g., GPU hours or FLOPs)? This information would help readers better evaluate the efficiency and scalability of the proposed method compared to existing pre-training approaches.
* The paper does not provide an implementation or anonymous code repository. Since reproducibility is crucial for evaluating new methods, could the authors consider releasing an anonymous version of the source code during the review phase? Without access to the implementation details, it may be difficult for others to fully reproduce the reported results.

**Limitations:**

No. The paper does not provide an impact or limitations statement. Including a brief discussion of the limitations of the proposed method and any potential societal implications would improve the completeness of the paper.

**Strengths And Weaknesses:**

***Strengths***
* This paper proposes a pre-training framework that utilizes an energy sentinel mechanism to optimize molecular-specific noise perturbation schemes during the learning process.
* The work employs the BRICS algorithm to decompose molecules into chemically meaningful motifs. It then computes intra-motif and inter-motif influence matrices and embeds this structural information into 3D conformations for denoising-based pre-training.
* Experimental results show that MOES-Pred outperforms several baseline models, including both supervised and 3D pre-training approaches, on three widely used molecular property prediction datasets: MD22, ISO17, and QM9.
* The paper further provides an interpretability analysis of the pre-trained model, comparing different noise perturbation strategies and demonstrating the effectiveness of the proposed approach.

***Weaknesses***
* The paper repeatedly refers to using a Graph Neural Network (GNN) as the backbone model for the proposed pre-training framework. However, the specific 3D GNN architecture employed in the pre-training stage is not clearly specified or described in sufficient detail. Since different 3D GNN architectures (e.g., equivariant models such as PaiNN, SE(3)-Transformer, or TorchMD-Net) can significantly affect performance and representation learning, clearly identifying the backbone model and its configuration is important for reproducibility and fair comparison with prior work.
* Another concern arises from the results reported in Table 2. For several properties, such as HOMO, $\mu$, and $C_{v}$, the performance differences between MOES-Pred and competing methods appear to be very small. It would therefore be helpful if the authors could provide statistical significance analysis (e.g., standard deviation across multiple runs or significance tests) to demonstrate that the reported improvements are not due to random variation. In addition, the tables currently highlight only the best-performing results. For clearer comparison, it would be beneficial if the authors also underline the second-best results in Tables 1 and 2, which would make it easier for readers to assess the relative improvements of MOES-Pred over strong baselines.
* Another limitation of this work is that the paper does not report the computational cost of the pre-training stage for MOES-Pred. Since the proposed method relies on large-scale pre-training on the PCQM4Mv2 dataset (3.4M molecules), understanding the required computational resources (e.g., training time, GPU usage, or FLOPs) would help readers evaluate the practical efficiency and scalability of the approach.

---

> ### Author Rebuttal · Authors · 2026-03-31
>
> $\textbf{Response to Reviewer LJSQ:}$
>
> $\textbf{Q1: MOES-Pred Model Backbone}$
>
> We provide the complete architectural configuration and hyperparameters below to ensure full reproducibility.
> 1) Model backbone of MOES-Pred: MOES-Pred is built upon the TorchMD-Net equivariant Transformer framework (Tholke & Fabritiis,2022; Luo et al., 2023) Within this backbone, we integrate an Energy Sentinel, an energy-oriented neural scoring component embedded directly in the end-to-end noise addition–denoising loop (Ni et al., 2024). The sentinel shares atomic and motif representations with the backbone GNN and operates as follows: after the model denoises each candidate perturbation, the sentinel monitors coordinate reconstruction errors and evaluates the physical plausibility of the reconstructed conformations through a weighted potential energy function (Eq.13). Rather than relying on fixed human-designed priors, the sentinel's scoring criteria, including atomic importance weights $w_{i}$ (Eq.12) derived from the influence matrix S and the energy constraint $E_{weighted}$ (Eq.13), are grounded in the GNN's learned representations. As the backbone network improves during training, the sentinel's energy constraints co-evolve accordingly: in early stages, scoring emphasizes enforcing physical rationality of molecular conformations; as training progresses, the emphasis gradually shifts toward reconstruction accuracy. This tight coupling between the sentinel's filtering strategy and the model's convergence state enables progressively more precise noise selection without manual tuning.
> 2) Architectural configuration on TorchMD-Net: Concretely, we introduce four lightweight modifications on top of the TorchMD-Net backbone: (i) A BRICS motif aggregation layer after the atomic embedding stage (Eq.7); (ii) A long-range inter-motif interaction enhancement module guided by the influence matrix (Eqs.8–10); (iii) An Energy Sentinel that employs a neural prediction network to dynamically score and filter noise perturbation schemes using a physically weighted energy function (Eqs. 12–16); (iv) Periodic noise optimization every 2 epochs, with a learnable balancing coefficient β (Eq. 10) controlling the weight ratio between long-range motif features and local atomic features.
> 3) hyper-parameter settng of MOES-Pred on PCQM4MV2:
>
> | Hyper-parameter | Value | Hyper-parameter | Value | Hyper-parameter | Value |
> | --- | --- | --- | --- | --- | --- |
> | activation | silu | embedding_dimension | 256 | batch_size | 70 |
> | Learning_rate (lr) | 0.0004 | lr_schedule | cosine |Dihedral angle exploration step size | 0.004
> | lr_factor | 0.8 | lr_min | 1.0e-07 | lr_warmup_steps | 10000 |
> | lr_cosine_length | 400000 | # attention heads | 8 | Sentinel loss weight | 0.01 |
> | Sentinel scoring frequency | Every 2 epochs | #noise scheme K | 24 | #radial bases | 50 |
> | Gaussian noise exploration step size | 0.001 | Bond angle exploration step size | 0.002 |Training_epochs | 10 |
>
> $\textbf{Q2: Statistical Significance Analysis on MOES-Pred}$
>
> Thanks for your suggestion. We provide statistical significance analysis below and will underline the second-best results in Tables 1-2 in the revised manuscript.
> Here is the significance analysis on QM9 among properties including HOMO, μ, and Cv, so as to prove that the improvement of MOES-Pred is not due to randomness.
>
> | Molecular Property Prediction | HOMO | $\\mu$ | $\\mathbf{C_v}$ |
> | --- | --- | --- | --- |
> | Mean | 15.2488 | 0.01118 | 0.02124 |
> | Standard deviation | 0.0807 | 0.000295 | 0.000416 |
> | Confidence Intervals | 15.2488 $\\pm$ 0.1001 | 0.01118 $\\pm$0.00037 | 0.02124 $\\pm$0.00052 |
> | RSD | 0.53\% | 2.64\% | 1.96\% |
>
> Based on the above data, the HOMO task exhibits the smallest relative error, with an RSD of 0.53% indicating the highest computational stability. Compared to the HOMO task, the µ and Cᵥ tasks show slightly decreased data stability, with RSDs of 2.64% and 1.96%, respectively; however, these results remain statistically significant. These results confirm that the improvements are stable across runs and not attributable to random variation.
>
> $\textbf{Q3: Computational Overhead on Pre-training Stage of MOES-Pred}$
>
> | Method | training time | # GPUs | FLOPs |
> | --- | --- | --- | --- |
> | MOES-Pred | 34h46m | 1 RTX 4090 | 20.72 E FLOPs |
> | Frad | 25h10m | 1 RTX 4090 | 14.95 E FLOPs |
> | Coord | 28h47m | 1 RTX 4090 | 17.10 E FLOPs |
>
> $\textbf{Q4: Release on Source Code of MOES-Pred}$
>
> see source code at: https://anonymous.4open.science/r/MOES-Pred/
>
> $\textbf{Q5: Addressing Limitations Concern on MOES-Pred}$
>
> MOES-Pred primarily focuses on molecular property prediction tasks utilizing 3D structural data as input and cannot accommodate other data modalities such as SMILES strings or molecular graphs. Future work may explore integration with complementary pre-training approaches to develop models capable of handling multi-modal molecular data.

---

> > ### Author Rebuttal · Reviewer_LJSQ · 2026-04-02
> >
> > Thank you to the authors for the thoughtful rebuttal. I do not have further questions at this stage and appreciate the clarifications provided. While the rebuttal was helpful, my overall evaluation remains unchanged, and I therefore intend to keep my score as is. Additionally, MOES-Pred requires significantly longer training time (34h 46m), which further limits my ability to increase the score.

---

> > > ### Author Response · Authors · 2026-04-07
> > >
> > > Thank you so much for your appreciation. Your acknowledgement is our motivation to step forward. Now, please allow us to rebuttal on “significantly longer training time (34h 46m)” of our MOES-Pred. Thanks for your patient reading.
> > >
> > > $\textbf{Q1. Overall Training Time of Existing Works}$
> > >
> > > $\textbf{1) One-time Pre-train with Negligible Fine-tune Cost.}$
> > >
> > > In fact, for our proposed MOES-Pred, although its full-parameter pre-training on large-scale unlabeled molecular dataset indeed incurs computational overhead, yet most importantly, the pre-training of MOES-Pred is a one-time amortized cost (only conducted once) and its fine-tuning on various downstream molecular property prediction tasks only involves lightweight parameter updates on small-scale labeled datasets, with negligible computational costs.
> > >
> > > $\textbf{2) Affordable Complexities among the Counterparts. }$
> > >
> > > We provide a detailed comparison on pre-training time among different counterparts (e.g., JOAO, JOAOv2, Mole-BERT, MoleMCL[1] and Frad[2]) under the same hardware configuration $\textbf{(more details in Table 3 and Table 19 at supplementary material respectively in MoleMCL [1] and Frad [2]). }$
> > >
> > > $\textbf{3) Conclusion:  }$
> > >
> > >  According to the facts below, MOES-Pred involves pre-training cost within only 34 hours on a single RTX 4090 GPU, demonstrating competitive training efficiency and reasonable computational cost compared with the majority SOTA pre-training methods.
> > >
> > > |Method	|Training time	|Hardware Configuration|
> > > | --- | --- | --- |
> > > |MOES-Pred	|34h46m	|1 RTX 4090|
> > > Frad|	25h10m|	1 RTX 4090|
> > > Coord|	28h47m|	1 RTX 4090|
> > > JOAO	|32h|	1 NVIDIA A100|
> > > JOAO2|	30h|	1 NVIDIA A100|
> > > Mole-BERT|	29h	|1 NVIDIA A100|
> > > MoleMCL|	21h|	1 NVIDIA A100|
> > > 3D-EMGP|	12h|	4 NVIDIA V100|
> > > SE(3)-DDM|	3-24h	|20 NVIDIA V100|
> > > Transformer-M|	75h	|4 NVIDIA A100|
> > > MoleBLEND|	24h	|4 NVIDIA A100|
> > > Uni-MOL|	20h	|8 NVIDIA V100|
> > > MOL-AE	|48h|1 NVIDIA A100|
> > >
> > > [1] Zhang et al. MoleMCL: a multi-level contrastive learning framework for molecular pre-training. Bioinformatics. 2024
> > >
> > > [2] Ni, Y. et al. Pre-training with fractional denoising to enhance molecular property prediction. Nature Machine Intelligence, 2024.
> > >
> > > $\textbf{Q2: Trade-off between Computational Head and Improvement on MOES-Pred }$
> > >
> > > Now, we theoretically analyze the time complexity of MOES-Pred, including its main mechanisms: the influence matrix and energy sentinel.
> > >
> > > Frad baseline model:$O_{t} = T * D * N^2 * B$. Influence matrix construction:$O_{t} = T * D * N^2 +C * N$. Energy sentinel:$O_{t} = K * T * D * N^2  * B+K * N+KlogK$. MOES-Pred:$O_{t} \approx  1.3 * T * D * N^2 * B * N$.
> > >
> > > K is the number of candidate perturbation schemes, T is the number of iterations, D is the feature dimension, N is the number of atoms, B is the batch size, $N^2$ denotes the influence matrix computation.
> > >
> > > $\textbf{We summarize the average improvement of MOES-Pred over the Frad baseline:}$
> > >
> > > | Downstream tasks | Dataset | Improvement over Frad |
> > > | --- | --- | --- |
> > > | Force field prediction tasks | MD22/ISO17 | +9.43% |
> > > | Quantum chemical properties | QM9 | +4.10% |
> > >
> > > Time complexity grows approximately linearly with the number of candidate perturbation schemes. Additional space complexity is limited to storing influence matrices, keeping overall memory usage within acceptable bounds.

---

### Official Review · Reviewer_6Uw9 · 2026-03-09

**Soundness:** 2
**Presentation:** 2
**Significance:** 2
**Originality:** 2
**Overall Recommendation:** 3
**Confidence:** 3

**Summary:**

This paper proposes MOES-Pred, a novel denoising pre-training framework that employs an energy sentinel mechanism to dynamically adjust molecule-specific noise perturbations. This method is evaluated on force prediction and quantum chemical property prediction, achieving state-of-the-art performance on both force prediction tasks and downstream quantum chemical property predictions.

**Compliance With Llm Reviewing Policy:**

Affirmed.

**Key Questions For Authors:**

1.Please elaborate on the differences and innovative points between this method and the frag method.On page 4, line 184, it illustrates four components of noise. Please explain the differences from the traditional definition methods (such as the Frad method), and provide a detailed explanation of the differences between bending vibration and torsional vibration.

2.On page 5, line 272, the authors claim that MOES-Pred uses molecule-specific noise perturbation schemes. Relative to the Frag baseline, what exactly makes the perturbation molecule-specific? In particular, what mechanism (or selection criterion) causes different molecules to receive different perturbation schemes instead of a single uniform policy? On page 8, line 424, the authors mention a molecule-specific amplitude design. How is this molecule-specific amplitude realized in practice?

3.Please clarify how many candidate perturbation schemes K are generated per molecule. How are the “physically plausible ranges” for Δxstretch,Δxbend,Δxtorsiondetermined in practice (e.g., based on bond/angle/dihedral statistics, force-field constraints, or heuristic thresholds)? In addition, what is the exact iterative update/selection strategy in MOES-Pred—do you discard the worst-scoring schemes and resample new ones each round, use a bandit-style explore–exploit update, or follow another deterministic/stochastic policy? Please provide an explicit algorithmic procedure and hyperparameter settings to ensure reproducibility and fair comparison.

4.Equation (6) is described as inter-molecular influence, while the benchmark tasks appear to be largely single-molecule property/force prediction. In implementation, is the influence matrix/score computed (i) within a molecule between atoms/motifs (intra-molecular), (ii) across molecules within a batch (inter-molecular), or (iii) via another mechanism (e.g., across conformations/views)? This clarification is important for the conceptual consistency and interpretation of the motif module, and may affect my assessment of its soundness and contribution.

**Limitations:**

Yes

**Strengths And Weaknesses:**

###  Strengths
The model was evaluated in terms of force prediction on MD22 and ISO17, as well as property prediction on QM9. It also included ablation studies where key modules were removed (removing sentinel / removing structural enhancement / removing molecule-specific noise). The results showed consistent degradation, which is reasonable to some extent.


### Weaknesses
1. The logic and content of the article need to be improved. For instance, Figure 2 is rather confusing and has poor readability. Is the example diagram of intra-motif effects a different motif of the example molecule? Table 2 shows that PaiNN compared the two times.

2. There is no clear indication regarding the model's applicability.The "mechanistic validation" of the innovation points is currently rather superficial: apart from ablation studies, there is a lack of more detailed analysis (such as which molecular types/degree of flexibility/cyclic structures benefit more), which makes the boundaries of the innovation contribution unclear.

---

> ### Author Rebuttal · Authors · 2026-03-31
>
> $\textbf{Response to Reviewer 6Uw9:}$
>
> $\textbf{Q1: Concerns on MOES-Pred Logic and Content}$
>
> 1) MOES-Pred V.S. Frad: Main difference lies in the noise addition strategy. Frad adopts RN and VRN for noise perturbation, while in MOES-Pred, stretch, bend, and torsion can not only represent RN and VRN through linear combinations, but also generate perturbations unattainable by them. This demonstrates MOES-Pred perturbation strategy encompasses a broader and more expressive operational space (see Eq (11, 16) & Fig. 4).
> 2) Bending vibration involves in-plane deformation of the bond angle formed by three atoms, and after perturbation, the atoms remain in the same plane; Torsional perturbation involves rotation around the bond of the dihedral angle formed by four atoms, and after perturbation, the atoms twist relative to each other.
>
>
> $\textbf{Q2: Intensive Comparison on Innovation between MOES-Pred and Frad Paradigm}$
>
> 1) Two core mechanisms: First, we customize noise generation by decomposing noise into stretch/bend/torsion/Gaussian components, using binary switches ($a_{s}$, $a_{b}$, $a_{t}$ ∈ {0,1}, Eq. 11, with $a_{s}+a_{b}+a_{t}$≥1) to activate perturbation modes adapted to molecular rigidity/flexibility. Second, dynamic screening is performed by the energy sentinel, which scores each candidate via Eq. 16, incorporating atomic importance weights from the influence matrix (Eqs. 12-13). Thus, different molecules receive structurally different high-scoring schemes.
>
> 2) Motivation on mechanism: First mechanism (combinatorial generation by stretch, bend, torsion, Gaussian) ensures each molecule is exposed to a structurally diverse set of candidate perturbations. Second mechanism (energy sentinel selection) evaluates which candidates produce physically plausible reconstructions for that specific molecule, retaining only those that balance conformational coverage with chemical validity. see Q2 response to reviewer HHY5.
> 3) Molecule-specific amplitude realization: The molecule-specific amplitude realization points to energy sentinel mechanism (see Eq (13-16)), a neural-based energy-oriented scoring module using prediction network to estimate per-atom potential energies, computing weighted energy scores, where lower scores indicate more physically plausible perturbations. Thus, different molecules will be guided to seek their most proper noise perturbation (amplitude) schemes. See “Iterative Update Strategy” algorithm in Q3, with hyper-parameter setting. See code github link in Q4.
>
> $\textbf{Q3: Guarantee on Molecule-Specific Noise Perturbation Schemes}$
>
> 1) Selection on K & Range Determination: Guided by the commonly used range of 10-50 for conformation generation in the drug discovery literature (COVER, Hemmerich J, 2020), we selected K=24 through systematic hyperparameter comparison. The perturbation ranges for $Δx_{stretch}, Δx_{bend}, and Δx_{torsion} $ are determined by MMFF94 force-field constraints and distributional statistics from the QM9 and MD22 datasets.
> 2) Iterative Update Strategy in MOES-Pred:
>
> For every 2 epochs:
>
> &emsp;#1.Evaluate: Calculate energy sentinel scores $S_{K}$ for all schemes
>
> &emsp;#2.Rank: Sort in ascending order by $S_{K}$, the lower the better, in Eq (16)
>
> &emsp;#3.Truncation selection:
>
> &emsp;&emsp;Retain:$P_{elite}$={p|rank(p)≤K/2}(top 50%, the lowest scores as best).
>
> &emsp;&emsp;Discard:$P_{discard}$={p|rank(p)>K/2}(bottom 50%, the highest scores as worst)
>
> &emsp;#4.Local sampling (40%):
>
> &emsp;&emsp;For each p∈$P_{elite}$:
>
> &emsp;&emsp;Sample new scheme p' from the parameter neighborhood N(p) collect into $P_{new-local}$.
>
> &emsp;#5.Global random (10%):
>
> &emsp;&emsp;Randomly sample $P_{new-global}$ from the global space
>
> &emsp;#6.Recombination: P=$P_{elite}$∪$P_{new-local}$∪$P_{new-global}$
>
> &emsp;&emsp;Output: Final high-score scheme pool
>
> Hyper-parameters: see full hyper-parameters setting in Q1 response of reviewer LJSQ.
>
> $\textbf{Q4: Typos on Inter-Molecular Influence}$
>
> This is a terminological error in the manuscript, for Eq (6), it is $\textbf{NOT inter-molecular, BUT inter-motif influence. The whole paper has been motivated oriented to inter-motif. Here, the influence matrix has been computed also as inter-motif }$(See source code at:https://anonymous.4open.science/r/MOES-Pred/)
> 1) Methodology Evidence: (Page 3, Line 93; Fig. 2, 4th subplot) Atoms within a motif are strongly covalently coupled, dominating local electron distribution and bond energy; atoms between motifs are weakly coupled only via van der Waals forces and steric hindrance (no direct chemical bonds), with drastically weaker interactions.
> 2) Equation Demonstration: Eqs. (8-9-10) confirm that the weights wi in the weighted energy Eweighted reflect the intra-motif strong/inter-motif weak coupling pattern. Inter-molecular influences are physically unrelated to single-molecule denoising, and will invalidate the sentinel scoring function, leading to chemically implausible learned atomic correlations.

---

### Official Review · Reviewer_HHY5 · 2026-03-18

**Soundness:** 4
**Presentation:** 3
**Significance:** 2
**Originality:** 2
**Overall Recommendation:** 4
**Confidence:** 3

**Summary:**

This paper proposes MOES-Pred, a pretraining framework for learning 3D molecular representations. The key idea is to move beyond fixed noise schedules in denoising-based pretraining and instead introduce molecule-specific perturbation schemes, selected via an energy sentinel mechanism that balances reconstruction accuracy and physical plausibility. In addition, the method incorporates chemical priors through BRICS-based motif decomposition and an influence matrix to enhance long-range interactions in the learned representation.

**Compliance With Llm Reviewing Policy:**

Affirmed.

**Final Justification:**

all concerns are well addressed, recommanded for accept

**Key Questions For Authors:**

•	What is the computational overhead introduced by the influence matrix construction and energy sentinel mechanism（training time compare to baselines）? Is this approach worth the added complexity for these benefits?

•	Can the method scale to significantly larger molecules?

•	The current ablation study removes entire modules (e.g., w/o E, w/o B, w/o N). Could the authors further decompose the contributions, for example: 1）different types of perturbations (stretching vs. torsion vs. Gaussian), 2）remove the weighting scheme in the energy sentinel

**Limitations:**

The main limitation of MOES-Pred is that it relies on existing high-quality 3D conformations as input, rather than generating 3D molecular frameworks from scratch. In addition, the current evidence is primarily derived from standard benchmarks and is insufficient to fully demonstrate its broad generalisation capabilities in more realistic, complex, and out-of-distribution molecular scenarios. Furthermore, the reported results on MD22, ISO17, and QM9 are strong, the downstream evaluation covers only two task families: force prediction and chemical property prediction. But whether the benefits of the proposed pretraining strategy transfer to other important settings, such as binding affinity prediction, conformation generation, or more application-oriented drug discovery tasks.

**Strengths And Weaknesses:**

Strengths:
1)	Well-motivated problem formulation grounded in physics and chemistry. Molecule-specific noise design is conceptually meaningful. Energy-aware training objective and integration of chemical priors through motif-based representation are reasonable.
2)	Comprehensive empirical evaluation across multiple benchmarks and including ablation studies. Authors honest about evaluating both the strengths and weaknesses of their work.


Weaknesses:
1) While the overall system is well engineered many components (denoising pretraining, motif pooling, attention-style aggregation, energy-based regularization) are extensions or combinations of existing ideas.
2) Performing representation learning on existing 3D structures, rather than a general-purpose 3D generation framework.
3) The paper’s contribution lies more in integration and task-specific design than in fundamentally new modelling techniques.
4) Then is for the potential computational cost of influence matrix construction. In addition, this paper does not discuss scalability to larger molecules or datasets.

---

> ### Author Rebuttal · Authors · 2026-03-31
>
> $\textbf{Response to Reviewer HHY5:}$
>
> Grateful for your comments. MOES-Pred introduces a qualitatively different paradigm, treating noise design as a per-molecule optimization problem via energy-based selection (Eqs. 11-16). We have conducted further experiments to demonstrate MOES-Pred's potential to scale to larger molecules.
>
> $\textbf{Q1: Computational Overhead on MOES-Pred}$
>
> 1)  We appreciate this important question. We theoretically analyze the time complexity of MOES-Pred. Frad baseline model:$O_{t} = T * D * N^2 * B$. Influence matrix construction:$O_{t} = T * D * N^2 +C * N$. Energy sentinel:$O_{t} = K * T * D * N^2  * B+K * N+KlogK$. MOES-Pred:$O_{t} \approx  1.3 * T * D * N^2 * B * N$. K is the number of candidate perturbation schemes, T is the number of iterations, D is the feature dimension, N is the number of atoms, B is the batch size, $N^2$ denotes the influence matrix computation.
>
> | Downstream tasks | Dataset | Improvement over Frad |
> | --- | --- | --- |
> | Force field prediction tasks | MD22/ISO17 | +9.43% |
> | Quantum chemical properties | QM9 | +4.10% |
>
> Time complexity grows approximately linearly with the number of candidate perturbation schemes. Additional space complexity is limited to storing influence matrices, keeping overall memory usage within acceptable bounds.
>
> $\textbf{Q2: Scale to Significantly Larger Molecules}$
>
> Grateful for your comments. We address scalability from both theoretical and empirical perspectives.
>
> 1) Theoretical considerations. (i) Chemical priors. The BRICS decomposition used for chemical priors can identify functional domains in proteins and repeating units in polymers; (ii) Long-range interaction modeling. In Eqs. (8-10), the inter-motif influence matrix aggregates features across motifs via weighted summation; the number of motifs grows sub-linearly with atom count; the residual connection in Eq. (10) injects non-local motif information to capture interactions that distance-based models miss; (iii) Energy Sentinel. The weighted energy function in Eq. (13) sums per-atom contributions, so its cost scales linearly with N.
>
> 2) Empirical evidence. We evaluate on the MD22 benchmark, which contains larger molecular systems ranging from 42 to 370 atoms. In Table 1, MOES-Pred achieves the best results on 5 out of 7 MD22 subtasks, specifically: -Buckyball (180 atoms): MAE 0.421 vs. Frad-RN 0.452 and Coord 1.810 (6.9% and 76.7% improvement, respectively); -Double-walled nanotube (370 atoms): MAE 0.705 vs. Frad-RN 0.786 and Coord 2.261 (10.3% and 68.8% improvement); -AT-AT-CG-CG (nucleic acid tetramer): MAE 0.269 vs. Frad-RN 0.308 (12.6% improvement). We reproduce these results below for convenience:
>
>
>
>
>
> $\textbf{Q3. More Decomposition on Ablation Study of MOES-Pred}$
>
> We appreciate this suggestion. Note that the isotropic Gaussian noise component cannot be removed, as it is required for the theoretical equivalence between the denoising objective and force field learning (see Appendix A.1). We use: w/o S (removes stretching noise), w/o Bend (removes bending noise), w/o T (removes torsional noise); w/o W (removes the importance-weighting scheme in the energy sentinel).
>
> **More detailed decomposition on Ablation Study of MOES-Pred**
> | Ablation | $\\mu$ | $\\alpha$ | HOMO | LUMO | Gap | $\\mathbf{R^2}$ | ZPVE | $\\mathbf{U_0}$ | $\\mathbf{U}$ | $\\mathbf{H}$ | $\\mathbf{G}$ | $\\mathbf{C_V}$ |
> | --- | --- | --- | --- | --- | --- | --- | --- | --- | --- | --- | --- | --- |
> | w/o S | 0.013 | 0.045 | 15.8 | 15.1 | 28.5 | 0.392 | 1.48 | 5.77 | 5.63 | 6.08 | 6.41 | 0.022 |
> | w/o Bend | 0.012 | 0.045 | 16.1 | 15.0 | 28.6 | 0.386 | 1.46 | 5.83 | 5.50 | 5.93 | 6.36 | 0.022 |
> | w/o T | 0.013 | 0.047 | 16.4 | 14.8 | 29.3 | 0.399 | 1.45 | 5.89 | 5.53 | 6.12 | 6.55 | 0.023 |
> | w/o W | 0.018 | 0.055 | 16.3 | 16.1 | 31.6 | 0.488 | 1.50 | 6.02 | 6.82 | 6.76 | 7.89 | 0.026 |
> | MOES-Pred | 0.011 | 0.040 | 15.2 | 14.2 | 27.7 | 0.344 | 1.43 | 5.38 | 5.31 | 5.69 | 6.01 | 0.021 |
>
> (i) removing the importance-weighting scheme from the energy sentinel (w/o W) causes substantial performance degradation, losing the ability to impose stricter supervision on structurally and functionally critical positions. (ii) among the three vibrational noise types, removing torsional noise (w/o T) has a slightly larger impact than removing bending (w/o Bend) or stretching noise (w/o S), though the differences between the three are relatively modest. This can prove: torsional rotations are the primary driver of conformational change in flexible molecules, and their removal limits the model's ability to explore conformational space. (iii) removing stretching (w/o S) or bending noise (w/o Bend) primarily affects local structural fidelity, with comparatively smaller impact on global conformational coverage.

---

> > ### Author Rebuttal · Reviewer_HHY5 · 2026-04-05
> >
> > Thank you for the detailed and constructive responses.
> >
> >
> > I appreciate the additional clarifications and experiments provided in the rebuttal. Overall, my main concerns regarding computational overhead, scalability, and ablation analysis have been adequately addressed.
> >
> >
> > The provided theoretical complexity analysis and empirical discussion clarify that the additional cost is moderate and scales reasonably with the number of perturbations. The reported overhead appears justified given the performance gains. The combination of theoretical and empirical results on larger molecular systems (MD22 benchmark) provides convincing evidence that the method can extend beyond small molecules. In addition, the more fine-grained ablation is helpful.
> > At this point, I do not have further concerns regarding these aspects, and I will raise increase my score to 4.

---

> > > ### Author Response · Authors · 2026-04-06
> > >
> > > Thank you so much for your appreciation. Your acknowledgement is our motivation to step forward. We will continue to devote our effort to polish this manuscript, and conduct further research.
> > >
> > > Again. Thank you so much.
> > >
> > > By the way, sir, sorry to bother. The overall recommendation is still with 3 score, where the final justification has not been updated yet. Thank you so much for your support.

---

### Decision · Program_Chairs · 2026-04-30

**Decision:**

Accept (regular)

**Comment:**

The paper proposes a novel denoising pre-training framework that dynamically adjusts molecule-specific noise perturbations. This is specifcially achieved via a motif-based chemical prior. It is evaluated in atomic-level force prediction and molecular-level quantum chemical property prediction.

The reviewers final rating is slightly positive (2x weak accept, 1x weak reject) yet there remain concerns in presentation/soundess and originality/novelty for, respectively, two reviewers. Several points raised in the reviews were addressed in the rebuttal, including basics such as statistical significance. While I do think that the sample-specific noise represents enough novelty, I agree that the presentation and evaluation could me more convincing. There is one (really nice) example, but this considers a single molecule.

Side note: It is great that the code was provided in the rebuttal, but a README would have been helpful.